# Genome-Wide Identification of DOF Gene Family and the Mechanism Dissection of *SbDof21* Regulating Starch Biosynthesis in Sorghum

**DOI:** 10.3390/ijms232012152

**Published:** 2022-10-12

**Authors:** Qianlin Xiao, Tingting Liu, Min Ling, Qiannan Ma, Wan Cao, Fangyu Xing, Tianhui Huang, Yingyi Zhang, Hong Duan, Zhizhai Liu

**Affiliations:** College of Agronomy and Biotechnology, Southwest University, Chongqing 400716, China

**Keywords:** sorghum (*Sorghum bicolor* L.), *SbDof21*, starch biosynthesis, transcriptional regulation

## Abstract

Starch is one of the main utilization products of sorghum (*Sorghum bicolor* L.), the fifth largest cereal crop in the world. Up to now, the regulation mechanism of starch biosynthesis is rarely documented in sorghum. In the present study, we identified 30 genes encoding the C2-C2 zinc finger domain (DOF), with one to three exons in the sorghum genome. The DOF proteins of sorghum were divided into two types according to the results of sequence alignment and evolutionary analysis. Based on gene expressions and co-expression analysis, we identified a regulatory factor, *SbDof21*, that was located on chromosome 5. *SbDof21* contained two exons, encoding a 36.122 kD protein composed of 340 amino acids. *SbDof21* co-expressed with 15 genes involved in the sorghum starch biosynthesis pathway, and the Pearson correlation coefficients (PCCs) with 11 genes were greater than 0.9. The results of qRT-PCR assays indicated that *SbDof21* is highly expressed in sorghum grains, exhibiting low relative expression levels in the tissues of roots, stems and leaves. SbDOF21 presented as a typical DOF transcription factor (TF) that was localized to the nucleus and possessed transcriptional activation activity. Amino acids at positions 182–231 of SbDOF21 formed an important structure in its activation domain. The results of EMSA showed that SbDOF21 could bind to four tandem repeats of P-Box (TGTAAAG) motifs in vitro, such as its homologous proteins of ZmDOF36, OsPBF and TaPBF. Meanwhile, we also discovered that SbDOF21 could bind and transactivate *SbGBSSI*, a key gene in sorghum amylose biosynthesis. Collectively, the results of the present study suggest that SbDOF21 acts as an important regulator in sorghum starch biosynthesis, exhibiting potential values for the improvement of starch contents in sorghum.

## 1. Introduction

Starch, one of the most important carbohydrates in the world, is the main component of human food and an important raw material for industrial applications [1,2]. Human beings have currently achieved the artificial synthesis of starch from carbon dioxide [3], while the current main way to obtain starch for mankind still depends on plant photosynthesis.

In plants, starch biosynthesis occurs in the plastids and undergoes a series of complex and coordinated biochemical reactions catalyzed by multiple enzymes [1,4]. Among these enzymes, ADP-glucose pyrophosphorylase (AGPase) catalyzes the biosynthesis of adenosine diphosphate-glucose (ADPG) to provide direct precursors for starch biosynthesis, presenting as a key speed-limiting step during starch biosynthesis [5]. Starch synthase (SS), including granule-bound starch synthase (GBSS) and soluble starch synthesis (SSS), is mainly responsible for the extension of sugar chains [1,6]. GBSS functions in the synthesis of amylase, and its mutation can lead to waxy (amylose-free) grains in wheat [7,8]. SSS catalyzes linear chain elongation by adding glucose units provided by ADPG to the non-reducing end of the acceptor chains, which is important for the synthesis of amylopectin [9,10]. The starch-branching enzyme (SBE) cuts α-1,4-glucan chains and transfers the segments to glucosyl residue in the C6 position, which is important for catalyzing the formation of the branch linkage [11]. The starch-debranching enzyme (DBE) is involved in amylopectin synthesis for hydrolyzing the wrong branch linkages and also participates in the formation of starch granules and the degradation of starch [12,13,14]. Besides AGPase, SS, SBE and DBE, another enzyme of starch phosphorylase (SP) is reported to have a function in starch biosynthesis [15,16,17]. Meanwhile, all those functional enzymes involved in starch synthesis contain isozymes that are encoded by multiple genes and exhibit tissue-specific expression patterns, constituting a complex mechanism of starch biosynthesis in plants [1,4,6].

In addition, the complicated regulatory networks also play key roles in starch biosynthesis. For example, AGPase is regulated by allosteric effectors, 3-phosphoglycerate and Pi [5]; ISA1 and ISA2 can form heterodimers in rice [18] and *Arabidopsis* [19]; SS and SBE tend to function in the form of complexes [20,21,22]; and the phosphorylation of functional enzymes is also one of the important regulation patterns for starch biosynthesis [23,24]. Meanwhile, the gene-encoding enzymes related to starch biosynthesis are induced by signal molecules and are directly regulated by transcription factors (TFs) at the transcriptional level, which already becomes an important regulation mode of starch biosynthesis in plants [4,25]. For example, HvSUSIBA2 in barley can regulate the transcription of *Iso1* by binding to the sugar response element within the promoter [26]; OsbZIP58 can directly bind to the promoter regions of *OsAGPL3*, *Wx*, *OsSSIIa*, *SBE1*, *OsBEIIb* and *ISA2* and regulate their expression [27]. RSR1, identified by gene co-expression analysis, is a negative regulatory TF for starch biosynthesis in rice [28]; OsNAC20/26—highly expressed in rice grains—are also involved in the transcriptional regulation of starch biosynthesis [29]. In maize, TFs of ZmbZIP91 [30], ZmMYB14 [31], ZmNAC126 [32], ZmNAC128, ZmNAC130 [33], O2 and PBF [34] are also reported to regulate the transcription of maize starch biosynthesis-related genes, thereby affecting starch contents in kernels. However, the biosynthesis and regulation mechanisms of starch are rarely documented in sorghum grain.

DNA binding with the C2-C2 zinc finger domain (DOF) family, a group of plant-specific TFs, contains a conserved region of 50 amino acids with a C2-C2 finger domain at the N-terminal that recognizes the cis-element containing (A/T) AAAG core sequence [35,36,37] and prolamin box (P-Box) [38,39]. DOF TFs have been reported to regulate multiple aspects of plant development. In *Arabidopsis*, OBF binding protein 1 (OBP1) can regulate the transcription of *CycD3;3* and regulate the plant cell cycle [40]. *AtDof2.1*, a jasmonic acid (JA)-inducible gene, can significantly reduce the promotion of leaf senescence [41]. In rice, OsDOF12 and OsDOF4 have been reported to participate in the regulation of flowering [42], while OsDOF12 and OsDOF18 are related to the regulation of nitrogen absorption in rice roots [43]. Meanwhile, an endosperm-specific gene in maize, *ZmDof3*, regulates starch accumulation and aleurone development [44]; ZmDOF36 can directly regulate the transcription of *ZmAGPS1a*, *ZmAGPL1*, *ZmGBSSI*, *ZmSSIIa*, *ZmISA1* and *ZmISA3*, playing important roles in starch biosynthesis in maize kernels [45].

DOF TFs exhibit diverse and important functions in different plants, including the cereals of rice, maize and wheat, while few documents are focused on the DOF family of sorghum. Although Kushwaha and colleagues have preliminarily identified the DOF TFs in sorghum [46], the functional profiling of DOF TFs, especially their regulations in starch biosynthesis, still remains less dissection in sorghum. Here, we identified 30 proteins containing conserved DOF domains from the entire sorghum genome through sequence characterizing. Phylogenetic analysis was performed on the amino acids, and we divided all sorghum DOF TFs into two groups, i.e., Group A and B. *SbDof21*, a highly expressed DOF gene in grains, was selected for further dissection of transcriptional regulation to the starch biosynthesis genes in sorghum grains.

## 2. Results

### 2.1. Identification of Sorghum DOF Proteins

A total of 30 DOF proteins were finally obtained from the sorghum genome via BLASTP queries. To better distinguish the corresponding sorghum Dof genes (*SbDof*s), we temporarily named those coding genes from *SbDof1* to *SbDof30* according to their order on the corresponding chromosomes (Appendix A). Meanwhile, *SbDof5* and *SbDof30* are different from those reported by Kushwaha et al. [46], showing up as newly identified Dof genes in sorghum. Among these *SbDof*s, eight are located on chromosome 1 (Chr1), six on Chr3, three on each of Chr2, Chr4, Chr8 and Chr9, two on Chr6, while only one on both Chr5 and Chr7 (Appendix A, Figure 1a). The number of amino acids encoded by 30 *SbDof*s ranged from 168 aa (*SbDof15*) to 560 aa (*SbDof13*), with the molecular weight and isoelectric points (pI) ranging from 17.25 kDa (SbDOF15) to 61.40 kDa (SbDOF13) and 4.75 (SbDOF17) to 10.27 (SbDOF13), correspondingly (Appendix A). Subcellular localization analysis of all sorghum DOF proteins (SbDOFs) demonstrated that SbDOF5 located on chloroplast thylakoid membrane, SbDOF10 and SbDOF16 on the mitochondrion, SbDOF12, SbDOF15 and SbDOF26 on extracellular space, SbDOF23 on chloroplast, while the remaining 23 SbDOFs all located in the nucleus (Appendix A).

### 2.2. Phylogeny and Sequence Characteristics of Dofs in Sorghum

Neighbor-joining (NJ) phylogenetic trees were constructed to reveal the evolutionary relationships among the members of *SbDof*s. Thirty identified *SbDof*s were divided into two groups: Group A and Group B (Figure 1b), similar to the phylogenetic results of Maximum likelihood (ML) (Appendix A). Groups A and B were further divided into three (A1 to A3) and five (B1 to B5) subgroups, respectively (Figure 1b). *SbDof21* and *SbDof25* from subgroup B5 exhibited a relatively closer relationship and formed an independent clade in B5 (Figure 1b). Similar trends were also observed among the pairs of *SbDof20/22* in B5, *SbDof2/12* in B3, *SbDof17/29* in B2, *SbDof3/27* in B1 and *SbDof6/8* in A3 (Figure 1b).

The results of gene structure analysis showed that all *SbDof*s contained only one or two exons (CDSs) (Figure 1c). Among the divided groups and subgroups, *SbDof*s from subgroups of B1, B2 and B3 exhibited only one exon, while all *SbDof*s of B4 contained two exons, *SbDof10* in B5 has only one exon, *SbDof22* has three exons, and others possess two exons (Figure 1c). All *SbDof*s in A1 and A3 in Group A contain two exons, while those in A2 exhibit a single-exon structure (Figure 1c).

To reveal the sequence features of SbDOFs, we performed domain and motif analysis. The results showed that all the SbDOFs contain conserved zf-Dof domain at a position closer to their N-terminals. Moreover, SbDOF13 contains another PTZ00121 superfamily domain at the N-terminal close to the zf-Dof domain with an unknown function currently (Figure 2a). The amino acid (aa) sequence of the conserved zf-DOF domain was further analyzed by sequence alignment. Twenty-nine identified SbDOFs all contained two C2 domains, while SbDOF15 possessed only one C at the region of the first C2 (Figure 2b). Even though, SbDOF15 still formed its conservative zf-Dof domain.

A total of 15 conserved motifs were detected in the 30 SbDOFs, i.e., Motif1 to 15 (Figure 2c). The multilevel consensus sequence and logos of conserved motifs are shown in Appendix A. Among these motifs, Motif1 was conserved and existed in all SbDOFs, while the other 14 motifs (Motif2 to Motif15) were only partially detected among specific SbDOFs (Figure 2c). For example, Motif2/4/6 were common to proteins of SbDOF6/8/11/13/14, Motif2/6 co-existed at the C-terminal of these proteins and Motif4 was located at the N-terminal near Motif1 (Figure 2c). Motif3 was observed at the N-terminus of seven SbDOFs, while Motif5/7 were only detected in both SbDOF21 and SbDOF25 (Figure 2c).

### 2.3. SbDof21 Highly Expressed in Sorghum Grains and Co-Expressed with Maize Starch Biosynthesis-Related Genes

Based on previous studies, we dissected the expression patterns of *SbDof*s among different sorghum tissues [47]. The results showed that the expression patterns of *SbDof*s were divided into two major types (Type i and ii). Additionally, 14 genes were involved in Type i; these genes were specifically expressed in grains or other tissues, among which both *SbDof21* and *SbDof25* exhibited high expression levels in grains (Figure 3a). However, the 16 genes in type ii were basically expressed in all tissues and formed a non-specific expression group (Figure 3a).

In our previous study, genes related to sorghum starch biosynthesis can also be divided into two categories based on their expression patterns. Type I covered 15 genes that were almost all highly expressed in sorghum grains, while Type II contained the rest of the 12 genes that exhibited relatively low expression levels in all tissues [47]. Co-expression analysis based on the expression data of RNA-sequencing (RNA-Seq) revealed that *SbDof21* exhibited a similar expression pattern to sorghum starch biosynthesis-related genes, and its Pearson correlation coefficient (PCC) with 15 sorghum starch biosynthesis-related genes were greater than 0.5 (Figure 3b). The PCC between *SbDof21* and 11 sorghum starch biosynthesis-related genes (*SbAGPS1*, *SbAGPS2*, *SbAGPLS1*, *SbGBSSI*, *SbSSIIa*, *SbISA1*, *SbPUL*, *SbSBEI*, *SbSBEIIb*, *SbPHOL*, *SbBt1*) were even greater than 0.9 (Figure 3b).

We further investigated the expression pattern of *SbDof21* in multiple tissues of sorghum cultivar BTx623 through qRT-PCR. The results showed that the transcripts of *SbDof21* could be detected among all 12 tissues, while the relative expression levels of *SbDof21* were apparently in four grain-related tissues of Seed_10/15/20/25 DAP rather than the other eight tissues, even the grain tissue of Seed_5DAP (Figure 3c).

### 2.4. Cloning and Sequence Analysis of SbDOF21

The coding sequence (CDS) of *SbDof21* contained 1023 bp and encoded a 36.122 kDa protein that consisted of 340 amino acids (Figure 4a). The protein sequence contains one zf-DOF domain at the N terminus, indicating that the cloned *SbDof21* was definitely a TF belonging to the DOF family (Figure 4b). Additionally, the results showed that another DOF gene, i.e., *SbDof25*, exhibited as the homolog of *SbDof21* in the sorghum genome, with sequence similarity of their cDNA > 71%. Meanwhile, *SbDof25* is also highly expressed in sorghum grains (Figure 3a).

### 2.5. SbDOF21 Was a Nuclear Localization Protein with Activation Activity

The transactivation activity of SbDOF21 was detected via a GAL4-based Y2H system. The experiment vector pGBKT7-SbDOF21, positive control pGBKT7-ZmMYB14 [31], and negative control pGBKT7 were transformed into yeast strain AH109 and screened on SD/-Trp plates. The transactivation activity was determined in the colonies cultivated on the SD/-Trp-His-Ura plates containing *X-α-gal*. The yeast contains the pGBKT7-SbDOF21 and pGBKT7-ZmMYB14 that could degrade the *X-α-gal* and turn blue, which demonstrated that SbDOF21 exhibited self-activating trans-activity (Figure 5a). The truncated SbDOF21 protein was prepared by removing the N-terminal or C-terminal residues, and its self-activation activity was detected. The results showed that 49 amino acids between the positions of 182 and 231 of SbDOF21 were important components of the activation domain, and the N-terminal and C-terminal sequences of SbDOF21 acted as inhibitors of its self-activation activity (Figure 5b).

Nuclear localization signals decide the nuclear localization of the transcription factors. In the prediction of the protein location, SbDOF21 was located in the nucleus (Appendix A). We further verified the subcellular location of SbDOF21 via maize leaf protoplasts. The enhanced green fluorescent protein (eGFP) signals driven by the 35S promoter could be detected in the protoplast cytoplasm, nucleus and cell membrane, while the eGFP signals of fusion proteins with SbDOF21 were only detected in the nucleus (Figure 5c). These results indicated that SbDOF21 was localized in the nucleus.

### 2.6. Binding Site Detection of SbDOF21 in the Promoter of Sorghum Starch Synthesis-Related Genes

Previous studies reported that PBF protein, DOF TF, could bind to P-box (TGTAAAG) and regulate the biosynthesis of grain protein and starch [44,48]. The results of sequence alignment and phylogenetic analysis showed that SbDOF21 and ZmDOF36 [45], ZmPBF1 [34], OsPBF [39] and TaPBF [38] exhibited homologous protein relationships (Figure 6a), and similar trends were observed within the phylogenetic results of Maximum likelihood (ML) (Appendix A).

In order to further reveal whether there are DOF protein binding sites in the promoter regions of sorghum starch biosynthesis-related genes, we intercepted 2000 bp sequences upstream of the transcription start sites (TSS) of 12 genes highly expressed in grains for binding motif analysis. The results turned out that different numbers of P-Box (TGTAAAG), (T/A)AAAG, GTAAAG and TGTAAAG motifs were detected with the 2000 bp TSS regions of 12 genes (Figure 6b). Among them, the 2000 bp TSS regions of *SbGBSSI* and *SbSBEIIb* contain seven related motifs, *SbBt1* contains nine and other sequences contain more than ten related motifs. These motifs make it possible for SbDOF21 to bind to the promoters and regulate the corresponding genes during starch biosynthesis in sorghum grains.

### 2.7. SbDOF21 Could Bind to the P-Box (TGTAAAG) Repeats In Vitro and Transactivate the Promoter of SbGBSSI

To identify whether SbDOF21 can bind with P-Box, a DNA fragment containing four tandem TGTAAAG repeats was used for the EMSA assay, and the recombinant protein of SbDOF21 was induced and purified from *E. coli*. The free probes were detected in three lanes, while shift bands were only detected in the right two lanes with the addition of SbDOF21 (Figure 7a). The addition of a 20-fold unlabeled probe can reduce the blocking band to a certain extent (Figure 7a). These results demonstrated that SbDOF21 could bind to the P-box in vitro.

We further cloned the promoter of *SbGBSSI* (1902 bp) with biological activity. Co-transformed the pUbi-Gus, pUbi-SbDof21 and pGreenII0800-pSbGBSSI-Luc into the maize leaf protoplast and detected the activities of Renilla (REN) luciferase and luciferase (LUC). The ratio of LUC/REN served as the standard to detect whether the SbDOF21 can activate or inhibit the activity of the promoter of *SbGBSSI*. The results showed that co-transforming SbDOF21 significantly increases the biological activity of the promoter of *SbGBSSI* (*p* < 0.01, Figure 7b). Moreover, SbDOF21 also presents similar transactivation to both *SbSSII* and *SbSBEIIb* (*p* < 0.05, Figure 7c). The results of the transient expression assays also suggest that SbDOF21 acts as a transcriptional repressor to the starch biosynthesis-related genes of *SbAGPLS1*, *SbAPGS2*, *SbAGPLS4* and *SbSBEI* but exhibits no transcriptional regulations to *SbSSI*, *SbISAII3* and *PHOL* (Figure 7c).

## 3. Discussion

*Sorghum bicolor* L. is an important C4 crop with high photosynthetic efficiency, which can be used to produce grain, feed, sugar/syrup and lignocellulosic biomass production for bioenergy [49]. Sorghum has the advantages of drought resistance, salt and alkali resistance and barren resistance, resulting in its extensive planting area, and has become the fifth cereal crop in the world [49,50]. Sorghum starch is also a carbohydrate used by human beings, which has important value in the feed industry and the brewing industry. However, there are relatively few studies on the molecular mechanism of sorghum starch synthesis [47]. Campbell and colleagues reported that the biosynthesis of sorghum starch depends on a series of functional enzymes, such as AGPase, SS, SBE and DBE [51]. These enzymes are also documented in maize, rice and other crops, and their functions have been studied [1,6,52]. The biosynthesis of starch is highly conserved in plants, and the research of other crops also provides a direction for the research of sorghum.

Previous studies document that there are different patterns in regulating the starch biosynthesis in crops, including allosteric effectors [5], protein interaction [18,20,22], protein phosphorylation [23,24], and transcriptional regulation [4,25], among which transcriptional regulation is popular and important for starch biosynthesis in plants. A large number of TFs have been reported to be involved in the transcriptional regulation of starch biosynthesis. For example, OsbZIP58 [27], OsNAC20/26 [29] and OsRSR1 [28] in rice; ZmbZIP91 [30], ZmMYB14 [31], ZmNAC126 [32], ZmNAC128/130 [33], O2 and PBF [34] in maize; HvSUSIBA2 in barley [26]; and TaNAC019 in wheat [53]. However, the transcription regulation in sorghum starch biosynthesis is rarely documented. In the present study, we reported that *SbDof21* is a grain highly expressed gene and is involved in the transcriptional regulation of starch biosynthesis in sorghum grains, which is consistent with other reported genes involved in starch biosynthesis [4].

DOF TFs family is plant-specific, and all DOF TFs possess conserved sequence features and played important roles in diverse directions of plant development, even in starch biosynthesis in maize [40,41,42,43,45,54]. In the present study, a total of 30 Dof genes, including two new members of *SbDof5* and *SbDof30*, were identified across the entire sorghum genome (Figure 1). The gene structure of *SbDof*s usually contains one to three exons (Figure 1c), similar to those reported in *Arabidopsis* and rice [35,46]. The exon number affects the diversity and the post-transcriptional processes of genes [55]. On the other hand, genes with low exon numbers are quickly induced and expressed in response to stimuli [55,56]. Conservative motif analysis found that there was a conservative motif, i.e., Motif1, contained by all SbDOFs, which is the motif that constitutes all the conservative domains of the sorghum DOF protein.

The sequence characteristics of DOF proteins are the basis of their biological function, and the diverse functions of DOF proteins have been reported in previous studies [40,42,43,48]. Currently, DOFs are believed to play key roles in the storage of protein biosynthesis and carbohydrate metabolism of endosperm and seeds of gramineous crops [34,38,39,44,45,57]. The results of the present study revealed that the DOF member of SbDOF21 tends to function similarly in starch biosynthesis in sorghum grains. We constructed the evolutionary tree of DOF proteins reported in rice [39], maize [44,45], wheat [38] and all DOF proteins in sorghum (Figure 6a). The results further confirmed their evolutionary conservatism and proved that the conserved function of SbDOF21 was consistent with the results of the present study that SbDOF21 was involved in the regulation of grain starch biosynthesis. We also found that there are motifs of P-box or (T/A) AAAG in the upstream sequence of starch biosynthesis-related genes in sorghum (Figure 6b), and SbDOF21 can directly bind to this tandem repeated P-box in vitro (Figure 7a).

Furthermore, SbDOF21 exhibited transactivation to the promoter of *SbGBSSI* in maize leaf protoplast (Figure 7b), which indicates that SbDOF21 potentially possesses the transcriptional regulating activity to the expression of *SbGBSSI*, similar to its ortholog of *Zm**Dof36* during 10DAP to 20DAP in developing maize kernels [45]. Moreover, *SbDof21* also exhibits diverse potential functions to the other starch biosynthesis-related genes in sorghum grains. For example, *SbDof21* can transactivate the expression of *SbSSII* and *SbSBEIIb* rather than *SbGBSSI* and can suppress the expression of *SbAGPLS1/3*, *SbSGPS2* and *SbSBE1* while presenting no significant functions to some other genes (Figure 7c). In the transient expression assays, Wu and colleagues also observed the transactivation of ZmDOF36 to *ZmISA1* [45], but in the present study, no significant activation or suppression was detected for SbDOF21 to the expression of both *SbISA1* (*p* = 0.0732) and *SbISA3* (*p* = 0.3924, Figure 7c). ISA is one of the isoenzymes of DBE that functions in the synthesis of amylopectin [12,13,14]. The different activation patterns between SbDOF21 and ZmDOF36 to *ISA*s might suggest the specific transcriptional regulations of SbDOFs to starch biosynthesis in sorghum grains. In conclusion, the results of the present study suggest that SbDOF21 is a candidate vital regulator of starch biosynthesis in sorghum grain.

## 4. Materials and Methods

### 4.1. Plant Materials

BTx623, provided by Rice and Sorghum Institute, Sichuan Academy of Agricultural Sciences (Luzhou, Sichuan, China), was grown under natural conditions on the college farm (College of Agronomy and Biotechnology, Southwest University, Chongqing, China). The tissues/samples of roots, stems, leaves, inflorescence and seeds were collected at different development stages, including jointing, flowering and maturity. All collected fresh tissues/samples were immediately immersed in liquid nitrogen and then stored at −80 °C for gene cloning and expression analysis. Three biological replicates were prepared for each sample.

### 4.2. Identification of Sorghum DOF Proteins

To identify all putative DOF proteins across the sorghum genome, we performed BLASTP (E-values < 1.0) queries via the NCBIv3 sorghum genome sequence of the Gramene database (http://www.gramene.org/, 26 March 2022) and NCBI (https://www.ncbi.nlm.nih.gov/, 20 December 2019), and the documented DOF protein sequences from *Arabidopsis*, rice and sorghum were used as the references [35,46]. ClustalW1.83 was used for multiple candidate sequence alignment to remove redundant and incomplete proteins. The conserved domain database (https://www.ncbi.nlm.nih.gov/cdd, 26 March 2020) was applied to confirm each protein containing the conserved region of 50 amino acids with a C2-C2 finger domain. Phylogenetic trees were constructed by using MEGA 5.10 via the neighbor-joining (NJ) method combined with a passion model in contrast to the maximum likelihood method phylogeny reconstruction results, and the bootstrap replicates were set to 1000 [58]. The length (no. of amino acids), pI and Mw of the sorghum candidate DOF proteins were summarized via Expasy (http://web.expasy.org/compute_pi/, 13 April 2021).

### 4.3. Sequence Characterizing of Sorghum DOF Proteins

MEME (http://meme.nbcr.net/meme/cgi-bin/meme.cgi, Version 5.4.1, 28 August 2022) was used to investigate the conserved motifs of sorghum DOF proteins. The minimum motif width, maximum motif width, and no. of different motifs were specified as 6, 50 and 20, respectively [46]. Gene Structure Display Server (GSDS2.0, http://gsds.cbi.pku.edu.cn/, 15 April 2015) was used to define the gene structure [59]. The intron distribution pattern and intron/exon boundaries of the candidate genes were obtained, and the displayed results were aligned with the 5′ terminal. Putative sorghum DOF protein nuclear localization signals (NLSs) were predicted via PSORT II (http://psort.ims.u-tokyo.ac.jp/, 28 August 2022).

### 4.4. Cloning and Expression Analysis of SbDof21

Total RNAs were isolated from different tissues of BTx623 through RNA Extraction Kit (Tiangen, Beijing, China) according to the manufacturer’s instructions. The quality and quantity of total RNA were verified via Nano Drop 1000 spectrophotometer and agarose gel. The PrimeScript^TM^ RT reagent kit with a gDNA Eraser (TaKaRa, Dalian, China) was used to obtain the first-strand cDNA and used for gene cloning and expression analysis. KOD enzymes (Toyobo, Osaka, Japan) with high fidelity were used to clone *SbDof21*. The amplified *SbDof21* products were constructed into the pMD-19T vector (TaKaRa, Dalian, China) and further verified by sequencing.

Quantitative real-time PCR (qRT-PCR) was performed via the Bio-Rad CFX96 real-time system in a total reaction volume of 10 µL Hieff qPCR SYBR Green Master Mix (Yeasen, Shanghai, China). The sorghum *eukaryotic translation initiation factor 4α* (*SbEif4α, SORBI_3004G039400*) was used as the internal control. All experiments were conducted four times, with three samples taken at each developmental stage. The relative transcription levels were calculated via the 2^-ΔΔCT^ method. The primers used for real-time PCR of *SbDof21* are named SbDof21QF (forward primer) and SbDof21QR (reverse primer) (Appendix A).

The expression pattern analysis of all sorghum DOF protein-coding genes is based on the RNA-sequencing (RNA-Seq) results of different tissues in our previous work [47], and TBtools is used for the heatmap drawing [60]. The co-expression analysis of *SbDof21* and sorghum starch biosynthesis-related genes was performed according to the Fragments Per Kilobase of exon model per Million mapped fragments (FPKM) of RNA-Seq data, and Pearson correlation coefficients (PCCs) were calculated with Microsoft-Excel 2016.

### 4.5. Functional Profiling of SbDof21 Gene

GAL4 two-Hybrid Yeast system was applied to study the self-activity of the transcription factor. The *SbDof21* and fragments deletion of *SbDof21* were sub-cloned into the pGBKT7 vector by using the sense primer with *Nde* I and the anti-sense primer with *BamH* I (Appendix A). The completed carrier pGBKT7-SbDOF21 was transformed into a yeast strain, AH109, to detect the activation of the transcription factor. SD/-Trp plates were used for positive screening, and the transformants were grown for three days under dark conditions of 28 °C. The monoclones were picked into 2 mL microtubes with liquid culture for propagation. All the monoclones, including positive and negative controls, were screened on SD/-Trp-His-Ura plates with *X-**α-gal* cultivated under dark conditions of 28 °C for three days to test the transcription activation.

The sub-cellular localization of SbDOF21 was analyzed by the transient expression of a fusion construct containing eGFP in the protoplasts of maize leaves. The extraction of protoplasts from maize leaves depends on the lysate liquid system constructed by 0.5 M mannitol, 1.5% cellulose, 0.5% macerozyre-R10, 10 mM EMS, 10 mM CaCl_2_ and 0.1% BSA. The constructed vector was transformed into protoplast through the PEG-Ca^2+^ method, and the transformed protoplasts were cultured in the dark for 16 h for fluorescence detection. The subcellular localization of eGFP and fusion proteins was detected under blue excitation light at 488 nm by a fluorescence microscope LSM 800 with Airyscan (Zeiss, Jena, Germany).

### 4.6. Over-Expression of Recombinant SbDOF21 Protein in E. coli

*SbDof21* was sub-cloned into pET32a (Takara, Dalian, China) vector for the prokaryotic expression. *BamH* I and *Hind* III were the restriction sites for vector construction. The primers are listed in Appendix A. Transetta (DE3) (Transgen Biotech, Beijing, China) was used as the host cell for prokaryotic expression. When the OD_600_ value of the propagation bacterial was 0.6, the isopropyl β-d-1-Thiogalactopyranoside (IPTG) was added for induction, and the final concentration of the inducer IPTG was 0.5 mM. The strains were continued to be cultured at 16 °C and 120 rpm overnight. The strains were collected by centrifugation and were broken discontinuously by ultrasound under 120 W for 10 min. The purification of the recombinant protein was according to the instruction of the Ni-Agarose His label Kit (CWBIO, Beijing, China).

### 4.7. Electrophoretic Mobility Shift Assay (EMSA) Assay

The SbDOF21-His recombinant protein was purified from the prokaryotic expression system. The single-stranded oligonucleotides 5′-TGTAAAGTGTAAAGTGTAAAGTGTAAAG-3′ and reverse complementary sequence 5′-CTTTACACTTTACACTTTACACTTTACA-3′ contained 4 P-Box (TGTAAAG) sequences to detect whether SbDOF21 could bind to the P-Box. The 5′ Biotin marker is directly added by the company (Sangon Biotech, Shanghai, China) in the primer synthesis. The conjugation reaction, competitive reaction and detection of electrophoretic mobility shift assay (EMSA) were performed according to the instruction of the Chemiluminescent EMSA Kit (Beyotime Biotechnology, Shanghai, China).

### 4.8. Dual-Luciferase Assay in Maize Leaf Protoplast

The pUbi-SbDof21:pGreenII0800-pSbGBSSI-Luc (1:2) was the experimental group, and pUbi-Gus:pGreenII0800-pSbGBSSI-Luc (1:2) was set as the control group. All the constructs were transformed into the maize protoplast through the PEG-Ca^2+^ method. Luciferase (LUC) and Renilla (REN) luciferase activities were measured via the Dual Luciferase Assay Kit (Promega, Madison, WI, USA) and analyzed via GloMax_2020 (Thermo Fisher Scientific, Waltham, MA, USA). LUC/REN ratio was calculated to measure the relationship between the experimental and control groups. Six independent experiments were performed, and each independent experiment consisted of three replicates. The difference with SbDof21 on the promoter activity of *SbGBSSI* was tested by *t*-test.

### 4.9. Sorghum Leaf Protoplast Transformation

Cellulase R-10 (Yakult, Japan) and Macerozyme R-10 (Yakult, Japan) were used to obtain sorghum leaf protoplasts. The enzymolysis liquid consisted of 1.5% Cellulase R-10, 0.5% Macerozyme R-10, 0.5 M mannitol, 10 mM 2-Morpholinoethanesulphonic acid monohydrate (MES monohydrate), 10 mM CaCl_2_ and 0.1% Bovine serum albumin (BSA). The cut sorghum leaves are immersed in the enzymolysis liquid and cracked at 25 °C at 50 rpm for 4 h. W5 buffer containing 2 mM MES, 154 mM NaCl, 125 mM CaCl_2_ and 5 mM KCl was used for sorghum leaf protoplast washing. MMG buffer containing 15 mM MgCl_2_, 0.4 M mannitol and 4 mM CaCl_2_ was used for sorghum leaf protoplast suspension and transformation. The transformation solution of protoplast was composed of 40% PEG4000, 0.8 M mannitol and 1 M CaCl_2_. The transformed protoplasts were cultured in a dark environment at 25 °C for 24 h. The cultured protoplasts were used for RNA extraction and gene expression detection. The qRT-PCR possessed three technical replications, and the *p*-value was calculated by *t*-test.

## 5. Conclusions

In this study, 30 genes encoding the C2-C2 zinc finger domain (DOF) were identified in the sorghum genome, including two genes of *SbDof21* and *SbDof25* that are highly expressed in sorghum grains. The co-expression analysis of *SbDof21* and starch biosynthesis-related genes confirmed *SbDof21* as a candidate regulator during starch biosynthesis. SbDOF21 is a typical DOF transcription factor that is localized to the nucleus and possesses transcriptional activation activity. Amino acids at positions 182–231 of SbDOF21 formed an important activation domain. P-Box and other DOF protein binding sites are generally observed within the sequences of 2000 bp upstream of the translation start site of sorghum starch biosynthesis-related genes. SbDOF21 can bind to four tandem repeat P-Box (TGTAAAG) sequences in vitro and regulate the transcription of these genes. Meanwhile, we found that SbDOF21 could transactivate *SbGBSSI*, a key gene in sorghum amylose biosynthesis. In summary, the results of the present study indicated that SbDOF21 acts as an important regulator in sorghum starch biosynthesis and exhibits potential values for the improvement of starch contents in sorghum.

## Figures and Tables

**Figure 1 ijms-23-12152-f001:**
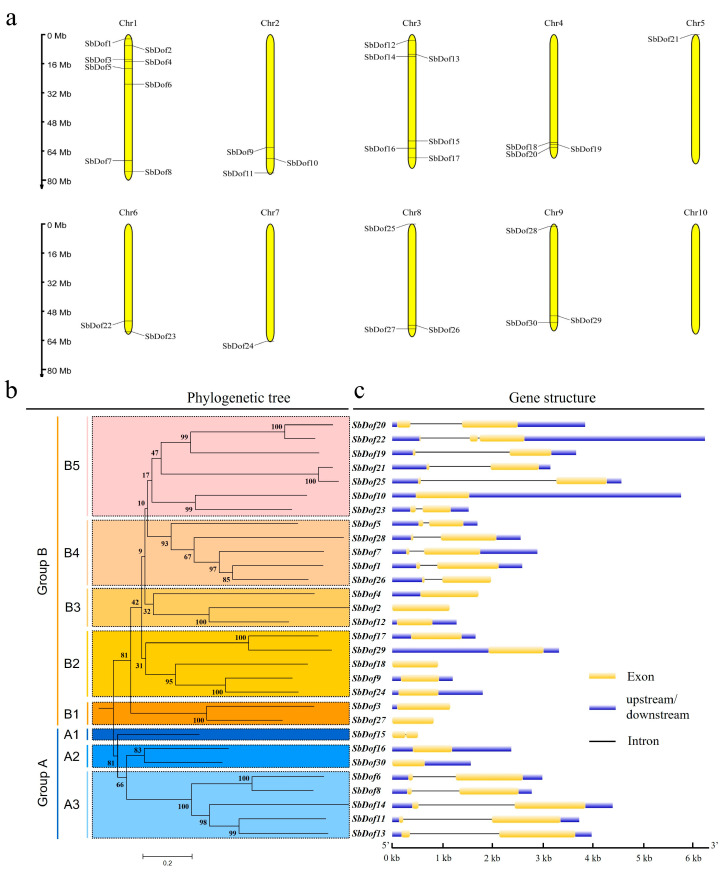
Characterizing 30 identified *SbDof*s across the sorghum genome: (**a**) Chromosomal localization of 30 identified *SbDof*s, the chromosome size is indicated by the left side rulers; (**b**) Phylogenetic analysis of 30 *SbD**of*s, A and B refer to two groups of *SbDof*s, while A1 to A3 and B1 to B5 to the corresponding subgroups of Group A and B, figures on each clade present the percentage resulted from the bootstrap analysis; (**c**) Gene structures of 30 *SbDof*s in sorghum.

**Figure 2 ijms-23-12152-f002:**
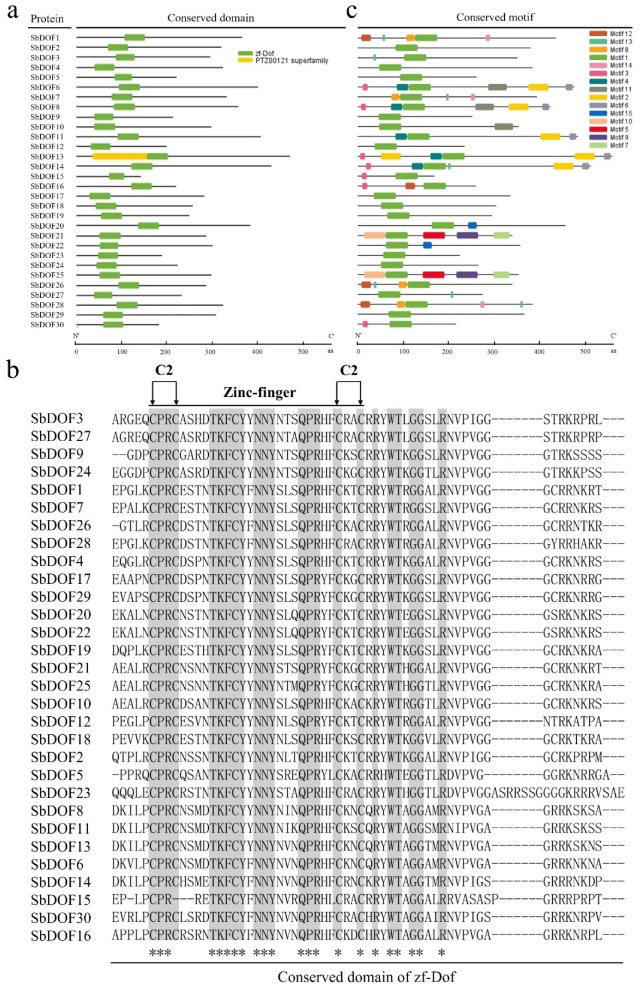
Sequence analysis of 30 DOF proteins in sorghum. (**a**) Conserved domain of 30 SbDOFs. The green box indicates the zf-DOF domain, and the lines represent amino acid sequences. (**b**) Sequence alignment of conserved domains of 30 SbDOFs. Gray labelled letters indicate the same amino acid sites in all proteins, * refers to the conserved amino acids among different SbDOFs. (**c**) Conserved motifs of zf-DOF among 30 identified DOF proteins in sorghum. Different colored boxes represent different motifs, and black lines represent amino acid sequences.

**Figure 3 ijms-23-12152-f003:**
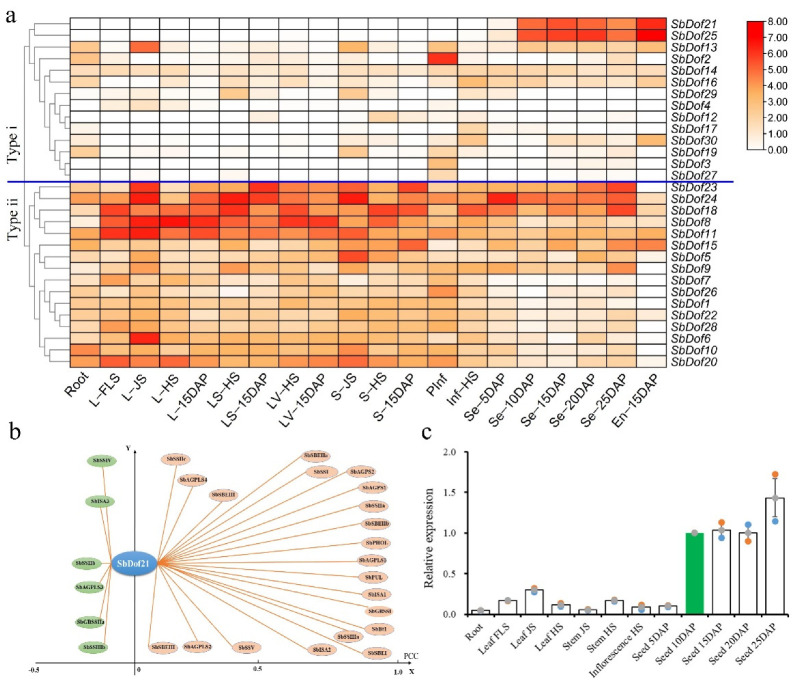
Expression profiling of *SbDof*s. (**a**) Expression data of 30 *SbDof*s in different tissues. (**b**) Co-expression analysis of *SbDof21* and sorghum starch biosynthesis-related genes. The vertical distance from a gene to the Y axis indicates the size of the Pearson correlation coefficient (PCC). (**c**) The expression pattern of *SbDof21* revealed via qRT-PCR. The green bar represents the relative expression level of Seed_10DAP that is standard to 1.

**Figure 4 ijms-23-12152-f004:**
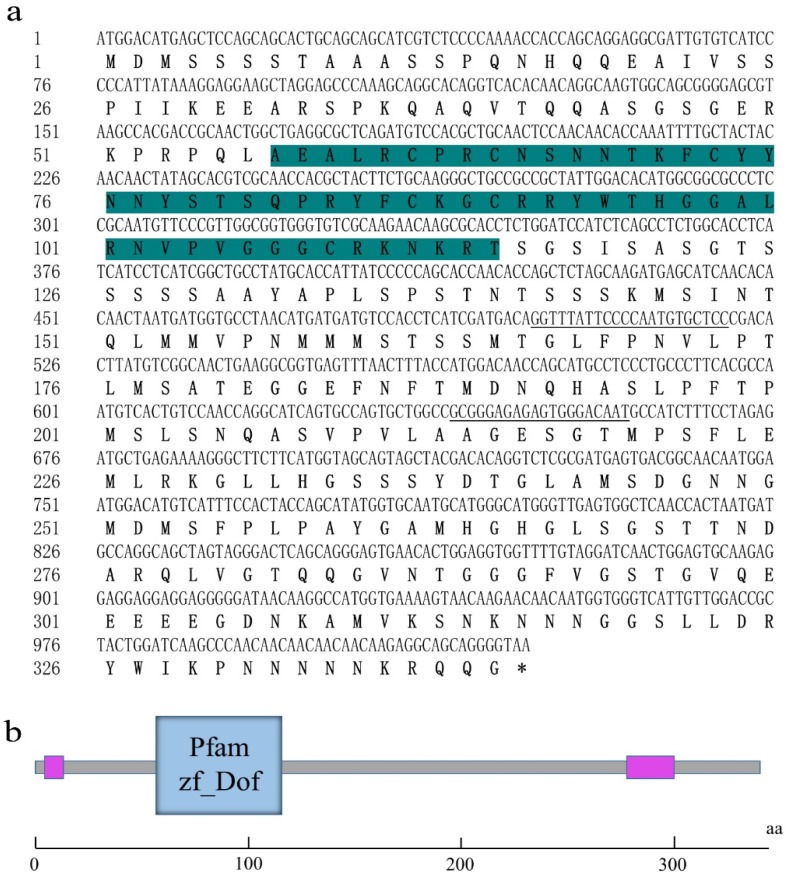
Coding Sequence (CDS) analysis of *SbDof21*. (**a**) The cloned CDS of *SbDof21* in sorghum cultivar (BTx623) and the corresponding coding products of amino acids. The blue background refers to the amino acids that formed the zf-DOF domain. The underlined DNA sequence refers to the primer position of qRT-PCR. (**b**) The predicted structure of the conserved domain of SbDOF21 by Smart. Pink boxes present low complexity regions.

**Figure 5 ijms-23-12152-f005:**
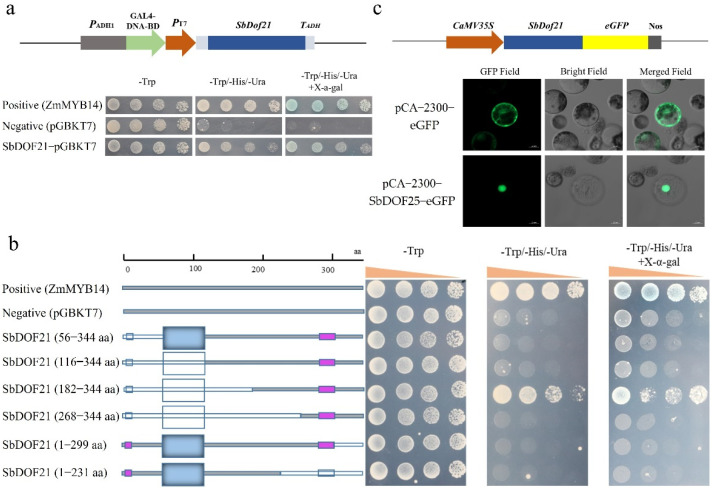
Functional characterization of SbDOF21. (**a**) Trans-activity assays of SbDOF21 in yeast strain AH109. ZmMYB14 with transactivation activity was used as a positive control [31]. Empty vector pGBKT7 was used as the negative control. (**b**) Transcriptional activation site identification of SbDOF21. The gray box referred to the amino acids, while the white box represented the excised sequence, pink box presented low complexity regions. Rectangular shapes refer to the zf-DOF domain. (**c**) The SbDOF21-eGFP fusion protein was driven by the 35S promoter and transiently expressed in the protoplasts of maize leaves. The eGFP driven by the 35S promoter transformed into the protoplasts of maize and was used as a control. GFP field, bright field and merged field indicate the state of fluorescent protein under three different channels.

**Figure 6 ijms-23-12152-f006:**
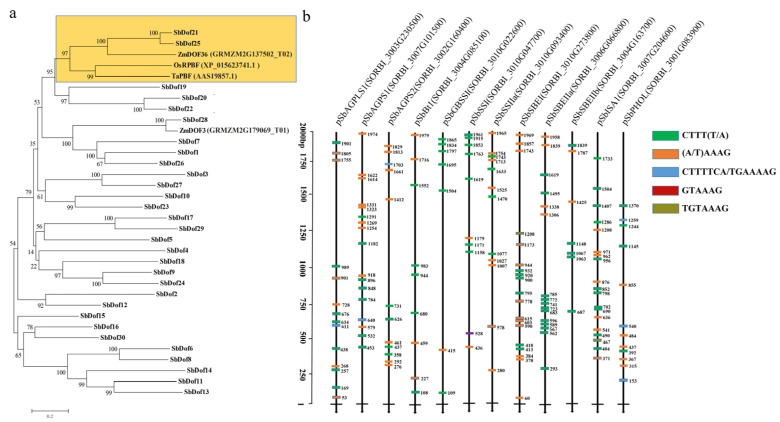
The orthologs analysis of SbDOF21 and P_Box and DOF protein binding sites were identified in the 2000 bp DNA sequence of sorghum starch synthesis-related genes. (**a**) Phylogenetic analysis of 30 SbDOFs, ZmDOF36 (GRMZM2G137502_T02), ZmDOF3 (GRMZM2G179069_T01), OsPBF (XP_015623741.1) and TaPBF (AAS19857.1). “Zm” means “*Zea mays* L.”, “Os” is the abbreviation of “*Oryza sativa* L.” and “Ta” represents “*Triticum aestivum* L.” (**b**) The P_Box and DOF protein binding sites were identified in the 2000 bp DNA sequence of 12 sorghum grains highly expressed starch synthesis genes from their TSS upstream. Different colored boxes represent different motifs, arrow lines represent the sequence position upstream of TSS and numbers represent the position of motifs in the sequence.

**Figure 7 ijms-23-12152-f007:**
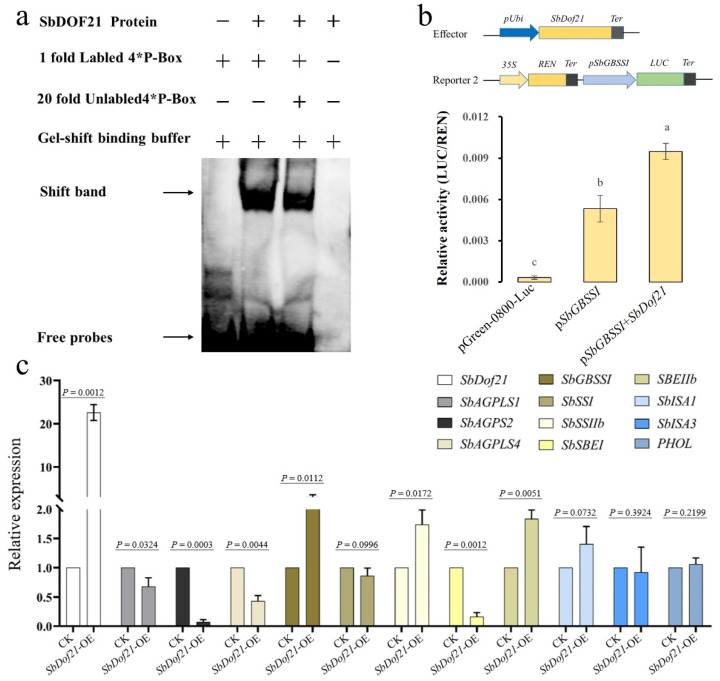
Study on the regulation mechanism of SbDOF21. (**a**) The binding results of SbDOF21 recombinant protein with the P-Box repeats in vitro. “+” represents the added composition and amount, and “-” means the corresponding component not added. (**b**) *SbDof21* affected the activities of *SbGBSSI* in transient expression assays. “a, b, c” are the results of the significance test at the level of *p* < 0.01. (**c**) Transient expression assays based on qRT-PCR of the functions of SbDOF21 to 11 starch biosynthesis-related genes in sorghum grains.

## Data Availability

Data are contained within the article or Appendix A.

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
