# Peer review of "Genome-Wide Identification of DOF Gene Family and the Mechanism Dissection of *SbDof21* Regulating Starch Biosynthesis in Sorghum"

_ijms, 2022, doi:10.3390/ijms232012152_

Round 1
Reviewer 1 Report
Reviewer (Comments for the Author):
Review of Genome-Wide Identification of DOF Gene Family and the Mechanism Dissection of SbDof21 Regulating Starch Biosynthesis in Sorghum
In this manuscript, the authors investigated the DOF gene family members in Sorghum using phylogenetic evolutionary analyses and sequence motif identification. Meanwhile, the authors identified SbDof21 as an important regulator during starch biosynthesis in Sorghum and performed interesting studies. However, I found several serious shortcomings that I think should be addressed by the authors.
SbDOF21 acts as an important regulator in sorghum starch biosynthesis, exhibiting potential values for the improvement of starch contents in sorghum. But the authors did not make the phenotypic data available to support this conclusion.
SbDOF21 and SbDOF25 relatively closer relationship and formed an independent clade in B5. Meanwhile, SbDof21 and SbDof25 exhibited high expression levels in grains. Why did the authors not mention whether the two genes cross functionally? I think it needs to be mentioned.
Line 77-78. In Arabidopsis, OBF binding protein 1 (OBP1) can regulate the transcription of CycD3; 3 and regulate plant cell cycle [40]. Indicate what 3 is.
Line 78. AtDof2.1, a jasmonic acid (JA)-inducible gene...Pay attention to the rules of gene and protein writing.
Line 82. Meanwhile, an endosperm-specific gene in maize, ZmDof3, regulates...Pay attention to the rules of gene and protein writing.
Line 170-172. In our previous study, the genes related to sorghum starch biosynthesis can also be divided into two categories based on their expression patterns, Type I covered 15 genes that almost all highly expressed in sorghum grains, while Type II contained the rest 12 genes that exhibited relatively low expression levels in all tissues [47]. The author described in the previous paragraph that Type I has 14 genes. How many genes are there in Type I?
Figure 3b. Co-expression analysis of SbDof21 and sorghum starch biosynthesis related genes. It's labeled SbDof25 instead of SbDof21.
Figure 4a. The cloned CDS of SbDof21 in sorghum cultivar (BTx623) and the corresponding coding products of amino acids. What does the blue background mean?
Figure 4b. The predicted structure of the conserved domain of SbDOF21 by Smart. What is the red box?
Figure 5a. The yeast contains the pGBKT7-SbDOF21 and pGBKT7-ZmMYB14 that could degrade the X-α-gal and turn blue, which demonstrated that SbDOF21 exhibited self-activating trans-activity (Figure 5a). I think this result is not obvious compared with the positive control.
Figure 6a. Include the Gene ID numbers when describing the genes. A few examples: ZmDOF36, ZmPBF1, OsPBF and TaPBF.
Figure 6b. Include the Gene ID numbers when describing the genes. Even make a Supplementary Table. In my experience, this can help a lot to the reader.
Figure 7a. The binding results of SbDOF21 recombinant protein with the P-Box repeats in vitro. The authors need more experiments to prove this conclusion. Meanwhile, authors should include mutations in the putative binding sites of P-Box (TGTAAAG) repeats to confirm the specificity of all the EMSA analysis.
Figure 7b. SbDOF21 affected the activities of SbGBSSI in transient expression assays. Based on the results in Figure 6b, there is no P-box (TGTAAAG) binding site on the 2000 bp sequences upstream of the transcription start sites (TSS) of SbGBSSI. Authors should find the binding site where SbDOF21 directly binds to the promoter of SbGBSSI.
Author Response
Author Responses:
Reviewer (Comments for the Author):1
Review of Genome-Wide Identification of DOF Gene Family and the Mechanism Dissection of SbDof21 Regulating Starch Biosynthesis in Sorghum
In this manuscript, the authors investigated the DOF gene family members in Sorghum using phylogenetic evolutionary analyses and sequence motif identification. Meanwhile, the authors identified SbDof21 as an important regulator during starch biosynthesis in Sorghum and performed interesting studies. However, I found several serious shortcomings that I think should be addressed by the authors.
SbDOF21 acts as an important regulator in sorghum starch biosynthesis, exhibiting potential values for the improvement of starch contents in sorghum. But the authors did not make the phenotypic data available to support this conclusion.
Response: Thanks a lot for the comments. Yes, we are also planning to perform more work focusing on the phenotypes, including the detecting of starch contents in sorghum grains. We deeply hope to get further directions when our following work submit to Int J Mol Sci.
SbDOF21 and SbDOF25 relatively closer relationship and formed an independent clade in B5. Meanwhile, SbDof21 and SbDof25 exhibited high expression levels in grains. Why did the authors not mention whether the two genes cross functionally? I think it needs to be mentioned.
Response: Thanks a lot for the directional comments. We are also conducting relevant research on the function of SbDof25, while up to now, we still do not collect enough sufficient evidence to reveal the crosstalk between SbDof21 and SbDof25. We hope to provide more genetic evidence to analyze the functional of both during our following work.
Line 77-78. In Arabidopsis, OBF binding protein 1 (OBP1) can regulate the transcription of CycD3;3 and regulate plant cell cycle [40]. Indicate what 3 is.
Response: Thanks for the comments. According to the Ref [40], “CycD3;3” stands together for a member of type D cyclin. We deleted the blank before the second “3”.
Line 78. AtDof2.1, a jasmonic acid (JA)-inducible gene...Pay attention to the rules of gene and protein writing.
Response: Revised. Thanks a lot for the detail and careful comments.
Line 82. Meanwhile, an endosperm-specific gene in maize, ZmDof3, regulates...Pay attention to the rules of gene and protein writing.
Response: Revised. Thanks a lot for the detail and careful comments.
Line 170-172. In our previous study, the genes related to sorghum starch biosynthesis can also be divided into two categories based on their expression patterns, Type I covered 15 genes that almost all highly expressed in sorghum grains, while Type II contained the rest 12 genes that exhibited relatively low expression levels in all tissues [47]. The author described in the previous paragraph that Type I has 14 genes. How many genes are there in Type I?
Response: Thanks a lot for the careful and directional comments. We are very sorry for the ambiguity here. “Type I” in “Line 170-172” indicates the starch biosynthesis related genes, while “Type I” in “Line159-160” is a classification of sorghum Dof gene according to expression patterns. We have revised accordingly.
Figure 3b. Co-expression analysis of SbDof21 and sorghum starch biosynthesis related genes. It's labeled SbDof25 instead of SbDof21.
Response: Revised. Thanks a lot for the comments.
Figure 4a. The cloned CDS of SbDof21 in sorghum cultivar (BTx623) and the corresponding coding products of amino acids. What does the blue background mean?
Response: Thanks a lot for the careful comments. We added the information of the blue background stands for in our revised version. Sorry for your carelessness.
Figure 4b. The predicted structure of the conserved domain of SbDOF21 by Smart. What is the red box?
Response: Thanks a lot for the careful comments. We added the information of the red boxes stand for in our revised version. Sorry for your carelessness.
Figure 5a. The yeast contains the pGBKT7-SbDOF21 and pGBKT7-ZmMYB14 that could degrade the X-α-gal and turn blue, which demonstrated that SbDOF21 exhibited self-activating trans-activity (Figure 5a). I think this result is not obvious compared with the positive control.
Response: Thanks a lot for the directional comments. We re-did the self-activating assays of SoDOF21, the revised results are more convincing than the previous. We also re-constructed Figure 5a accordingly.
Figure 6a. Include the Gene ID numbers when describing the genes. A few examples: ZmDOF36, ZmPBF1, OsPBF and TaPBF.
Response: Thanks a lot for the directional comments. Gene ID and accession number have been added accordingly.
Figure 6b. Include the Gene ID numbers when describing the genes. Even make a Supplementary Table. In my experience, this can help a lot to the reader.
Response: Thanks a lot for the directional comments. Gene ID and accession number have been added accordingly.
Figure 7a. The binding results of SbDOF21 recombinant protein with the P-Box repeats in vitro. The authors need more experiments to prove this conclusion. Meanwhile, authors should include mutations in the putative binding sites of P-Box (TGTAAAG) repeats to confirm the specificity of all the EMSA analysis.
Response: Thanks a lot for the directional comments.
We added the experiment of overexpression of SbDof21 in sorghum leaf protoplasts to detect the expression of starch biosynthesis related genes, including SbGBSSI. We analyzed the binding specificity of SbDOF21 in vitro, which can preliminarily indicate that it is similar to the homologous PBF and may participate in the transcriptional regulation of sorghum starch biosynthesis. The P-box repeat sequence is used in EMSA, and it may not be of practical significance to set the mutation. We will perform the EMSA analysis of this mutant P-box in the specific promoter.
Figure 7b. SbDOF21 affected the activities of SbGBSSI in transient expression assays. Based on the results in Figure 6b, there is no P-box (TGTAAAG) binding site on the 2000 bp sequences upstream of the transcription start sites (TSS) of SbGBSSI. Authors should find the binding site where SbDOF21 directly binds to the promoter of SbGBSSI.
Response: Thanks a lot for the directional comments.
There is no P-box (TGTAAAG) binding site within the region of 2000 bp upstream the transcription start sites (TSS) of SbGBSSI, but there is a binding motif of other DOF proteins. These observations also indicate that P-Box is not the only binding site of SbDof21. This finding also implies that detecting the binding sites and revealing the regulation mode of SbDof21 to SbGBSSI is important for the mechanism dissection of transcriptional regulation in sorghum starch biosynthesis.
Reviewer 2 Report
The author studied the structure and function of Dof gene family in Sorghum and they used the several tools and methods. I think that the structure of manuscript should be improved before publication. My comments include:
- Line 78, 82, etc.: Gene name should be provided in italic format. Please check the entire text.
- Neighbor-joining (NJ) method is not suitable for studying the evolution relationships. ML (maximum likelihood) method is recommended.
- I recommend to add orthologues of SbDofs from other plants such as rice, maize, etc. to phylogeny tree.
- To construct a phylogeny tree, the protein sequences are most used. So, protein name should not write in italic format.
- Sequence or logo of conserved motifs should be added to supplementary data.
- The discussion is not well written, the first few lines are more suitable for the introduction section. Key results should be interpreted.
- I suggest this sentences to line 293: The exon number affects the diversity and the post-transcriptional processes of genes (Koralewski and Krutovsky, 2017). On the other hand, genes with low exon number are quickly induced and expressed in response to stimuli (Koralewski and Krutovsky, 2017; Heidari et al, 2022).
- References:
Koralewski and Krutovsky, 2017: https://doi.org/10.1371/journal.pone.0018055
Heidari et al, 2022: https://doi.org/10.3390/agronomy12102253
Author Response
Author Responses:
Reviewer (Comments for the Author):2
The author studied the structure and function of Dof gene family in Sorghum and they used the several tools and methods. I think that the structure of manuscript should be improved before publication. My comments include:
- Line 78, 82, etc.: Gene name should be provided in italic format. Please check the entire text.
Response: Revised. Thanks a lot for the comments.
- Neighbor-joining (NJ) method is not suitable for studying the evolution relationships. ML (maximum likelihood) method is recommended.
Response: Thanks a lot for the comments. We provide the results of ML as the supplementary figure. We observed some slight differences between the NJ and ML, while both results of NJ and ML still stand for the trend of two major types, i.e. A and B.
- I recommend to add orthologues of SbDofs from other plants such as rice, maize, etc. to phylogeny tree.
Response: Thanks a lot for the comments. The main purpose of the present study is focusing on the functional mining of DOF homologous genes in sorghum. That’s why only SbDofs are covered. In addition, similar work as you mentioned already was documented by Noguero and colleagues (Noguero et al., 2013). We also quoted their findings in our present work.
Noguero M, Atif RM, Ochatt S, Thompson RD. The role of the DNA-binding One Zinc Finger (DOF) transcription factor family in plants. Plant Sci. 2013, 209:32-45. doi: 10.1016/j.plantsci.2013.03.016.
- To construct a phylogeny tree, the protein sequences are most used. So, protein name should not write in italic format.
Response: Revised. Thanks a lot for the comments.
- Sequence or logo of conserved motifs should be added to supplementary data.
Response: Revised accordingly. Thanks a lot for the comments.
- The discussion is not well written, the first few lines are more suitable for the introduction section. Key results should be interpreted.
Response: Revised accordingly. Thanks a lot for the comments.
- I suggest this sentences to line 293: The exon number affects the diversity and the post-transcriptional processes of genes (Koralewski and Krutovsky, 2017). On the other hand, genes with low exon number are quickly induced and expressed in response to stimuli (Koralewski and Krutovsky, 2017; Heidari et al, 2022).
- References:
Koralewski and Krutovsky, 2017: https://doi.org/10.1371/journal.pone.0018055
Heidari et al, 2022: https://doi.org/10.3390/agronomy12102253
Response: Revised accordingly. Thanks a lot for the comments.
Round 2
Reviewer 2 Report
Thanks.
The revised version is improved.
Please just update the list of supplementary files (line 509).